# OpenReview forum: "Benign Samples Matter! Fine-tuning On Outlier Benign Samples Severely Breaks Safety"
_ICML.cc/2025/Conference — ICML 2025 spotlightposter_

### Official Review · Reviewer_UXrg · 2025-02-23

**Overall Recommendation:** 4

**Summary:**

This paper is subsequent work following (He et al, 2024) .The paper proposes a better benign fine-tuning attack based on the influence function techniques.

**Claims And Evidence:**

Yes.

**Essential References Not Discussed:**

* A con-current work Virus[1]  is also focusing on harmful fine-tuning attack aiming to bypass guardrail moderation.

[1] Virus: Harmful Fine-tuning Attack for Large Language Models Bypassing Guardrail Moderation[J]. arXiv preprint arXiv:2501.17433, 2025.

I am aware that the authors have no obligation to discuss the con-current work. However, I would suggest the authors to add a section to discuss it  because they are actually aiming to solve the very same problem, but in two very different directions.

Specifically, OpenAI and other service provider recently adopt guardrail moderation to filter out harmful samples for their fine-tunig API. However, it is obvious that this ad-hoc solution is problematic, and there recently there is a rise of paper working on designing a better attack to it. The design directions mainly on three main directions.

1. How to make benign data **stronger** in attacking the safety aligned model?  (Qi et al,2023) first show that benign fine-tuning attack can also compromise safety. However, it is shown by subsequent work that such benign fine-tuning attack is not as successful as attack with harmful data.  (He et al, 2024) show that one can sample stronger “benign data” that can better attack the safety aligned model.

2. How to make harmful data **stealthier** to bypass the guardrail modeation? (Halawi et al,2024) is the first attempt working on this problem. However, the main weakness of this paper is that by adopting their attack, the users in the testing time need to cipher the harmful quesitons into human-unreadable text and decipher the harmful answer transmitted from the server. This paradign actually limits the use case of the harmful fine-tuning attack because the answers from the server is actually is not human readable harmful answers.  To address this issue,  Virus[1] is a subsequent attempt, which aim to jailbreak the guardrail detection and poison the victim model directly.

I believe there are more papers coming out alone these two lines of research, and it is necessary to explicitly inform the readers that there are two lines of research endeavorscurrent going on.

* Some other relevant work on the same topic are missing from discussion:

Lora fine-tuning efficiently undoes safety training in llama 2-chat 70b

 Identifying and Tuning Safety Neurons in Large Language Models

Targeted Vaccine: Safety Alignment for Large Language Models against Harmful Fine-Tuning via Layer-wise Perturbation

 Assessing the brittleness of safety alignment via pruning and low-rank modifications

Mitigating fine-tuning jailbreak attack with backdoor enhanced alignment

Do as I do (Safely): Mitigating Task-Specific Fine-tuning Risks in Large Language Models

Safety Layers in Aligned Large Language Models: The Key to LLM Security

 SEAL: Safety-enhanced Aligned LLM Fine-tuning via Bilevel Data Selection

Safety Alignment Shouldn't Be Complicated

SaLoRA: Safety-Alignment Preserved Low-Rank Adaptation

Towards Secure Tuning: Mitigating Security Risks Arising from Benign Instruction Fine-Tuning

Safety-Aware Fine-Tuning of Large Language Models

Defending Against Unforeseen Failure Modes with Latent Adversarial Training

A safety realignment framework via subspace-oriented model fusion for large language models

Safe lora: the silver lining of reducing safety risks when fine-tuning large language models

Locking Down the Finetuned LLMs Safety

Probe before You Talk: Towards Black-box Defense against Backdoor Unalignment for Large Language Models

Separate the Wheat from the Chaff: A Post-Hoc Approach to Safety Re-Alignment for Fine-Tuned Language Models

NLSR: Neuron-Level Safety Realignment of Large Language Models Against Harmful Fine-Tuning

Panacea: Mitigating Harmful Fine-tuning for Large Language Models via Post-fine-tuning Perturbation

On Weaponization-Resistant Large Language Models with Prospect Theoretic Alignment

No two devils alike: Unveiling distinct mechanisms of fine-tuning attacks

 Your Task May Vary: A Systematic Understanding of Alignment and Safety Degradation when Fine-tuning LLMs

On Evaluating the Durability of Safeguards for Open-Weight LLMs

Model Tampering Attacks Enable More Rigorous Evaluations of LLM Capabilities

Defending against Reverse Preference Attacks is Difficult

Emerging Safety Attack and Defense in Federated Instruction Tuning of Large Language Models

 PEFT-as-an-Attack! Jailbreaking Language Models during Federated Parameter-Efficient Fine-Tuning

**Experimental Designs Or Analyses:**

Yes. However, more results can be provided. See Weakness for details.

**Methods And Evaluation Criteria:**

Yes. Fine-tuning attack on benign data is making perfect sense because benign data are hard to be filtered out by Guardrail moderation. And  it also makes sense that a subset of benign data can compromise safety more serious, as it is not uncommon to see in other problems (tasks) that a smaller dataset works better than a larger one.

**Other Comments Or Suggestions:**

I suggest the authors to name "Mitigation against Malicious Fine-tuning"  in Page 3, line 130 to  "Mitigation against Harmful Fine-tuning", and also replace malicious fine-tuning with harmful fine-tuning in any other places of the paper.

My justification is as follows:

1. The name of harmful fine-tuning attack is first used in [2], first available in Feb 26, 2024. The name of "malicious fine-tuning" is first used by [3], first available in June 28, 2024.  I guess the concept that the authors want to express is exactly the same with the concept proposed in [2]. In terms of first appearance time to the public, the concept of harmful fine-tuning attack is earlier.

[2] Immunization against harmful fine-tuning attacks

[3] Covert malicious finetuning: Challenges in safeguarding llm adaptation

2. The concept of harmful fine-tuning attack is explicitly defined in Section 2 in [2]. In contrast, in [3], the concept of "fine-tuning threat model", instead of "malicious fine-tuning", is defined in their Section 2.

3. There are already a line of work using the name of harmful fine-tuning attack, and are accepted already (so they cannot change their titles). I list out a few as follows:

 Immunization against harmful fine-tuning attacks EMNLP2024

Vaccine: Perturbation-aware alignment for large language model aginst harmful fine-tuning NeurIPS2024

Lazy safety alignment for large language models against harmful fine-tuning NeurIPS2024

Representation noising effectively prevents harmful fine-tuning on LLMs NeurIPS2024

Booster: Tackling harmful fine-tuning for large language models via attenuating harmful perturbation ICLR2025

NLSR: Neuron-Level Safety Realignment of Large Language Models Against Harmful Fine-Tuning AAAI2025

**Other Strengths And Weaknesses:**

**Strength**
1. Paper is easy to read.
2. The performance is better than existing SOtA solution by (He et al, 2024)
3. The attack is tested against SOTA defense against harmful fine-tuning. The authors show that their attacks are effective even though defense such as safeInstrc or Lisa is adopted.  The evaluation is really impressive because some of the defense just appear very recently.
4. The code is well organized with runnable script.  This make it easier to reproduce the results.
5. The finding that short answer benign sampels sriously downgrade safety but fine-ting only on them compromise utility is novel. I also agree the authors explanation with shallow alignment hypothesis.

**Weaknes**
1.  Evaluation can be more comprehensive. As all the results in Table 1 are using GPT for evaluation, I suggest the authors to  add an experiment on GSM8k, which at least does not rely on GPT score to evaluate the utility.

2. The utility is downgraded very significantly after fine-tuning. This raised concern on whether the fine-tuning enable the models to effectively learn the downstream task.

3. The experimental setting is strange. From table 1, all the utility after fine-tuning is downgraded, even under full benign fine-tuning. This is in contrast of the goal of fine-tuning. We want the model to be better (in terms of at least a sub-task) instead of making the model performing worst. Therefore, I recommend the authors to experiment on GSM8K and plot the same information in Table 1. The utility of GSM8K should increase after fine-tuning on GSM8k samples.

4. The logistical flow of the attack method design is strange.  The authors seem to test the self-influence score  (Pruthi et al., 2020) in the benign fine-tuning problem context without much intuituion. Before experiment, it is unknown and without any intuition why the outlier samples actually compromise safety. After experiment, the authors do discover that the outlier samples are data with shortest answers, and subsequent associate the success of attack to shallow alignment hypothesis.  However, such a way to design method might be problematic and also risky.  As pointed out by my next question, it seems that the reason that the attack method work is just a random coincidence with other factors.

5. (Major!) A subsequent question is that: is the outlier score really useful? Let's say there exists another dataset with most data with short answer and few data with long answer. Based on the outlier score, it should be the long answer data being selected. Then will the attack method be useful? I think the answer is probably no, because  the authors already showcase that it is the short-toekn samples that is capable of degrading safety alignment of LLMs, but not the long one. If the answer is yes, then it actually contradict your own finding "Influence of Samples with Short Token Lengths." in Page 4.  I looks forward to more discussion on this issue. It seems that the finding and method design in this paper is contradicting each other.

6. It is not obvious how the proposed method advance existing method BA(He et al,2024). I only see two group of data in Table 1. One group of HS on Alpaca is even worst than BA.

7. (minor) The method seems to be a direct application of (Pruthi et al., 2020) without much modification to the current problem context, and therefore novelty is not that significant. I don't think this is very important for acceptance of the paper as long as the method is useful. However, I do think that this is an important criterion for a outstanding paper because a novel method can give insight and have more influence to other sub-fields, which apparantly this paper is lacking. I don't expect the authors to solve this concern but just want to point it out here.

**Questions For Authors:**

1. For evaluation on Lisa, are you using the Bianchi safety samples or the Bevertails original alignment samples?
I also observe a similar issue with Bevertails original alignment dataset --its alignment performance is simply not ideal. Later in the RepNoise paper,  (Rosati, 2024)  offers a better alignment dataset.  See the refusal column in https://huggingface.co/datasets/anonymous4486/repnoise_beavertail

2. In terms of the statement "bi-state optimization strategy has limited effectiveness in suppressing the harmfulness of the benign samples", I find that this statement contradict the results in the Lisa paper. In page 20, Table 15 of the Lisa paper, they do perform an exepriment against Bi-directional Anchoring (He et al., 2024) and their results show that Lisa is quite effective against fine-tuning with benign sampling, with or without advanced data selection of Bi-directional Anchoring. I conjecture the difference of results are due to some different experimental settings, but I need more investigation for figuring out the exact reason. I hope the authors can mention here that the evaluation results are different in the two papers. Future readers might be interested in this discrepancy.

3. I also appreciate if the authors can give some insights of why benign fine-tuning attack seems to be stronger in the testbed of (He et al,2024) (same testbed with this paper) but fails in Lisa's testbed. What is the key factor that enables/disables the success of benign fine-tuning attack?   My conjecture is that it is because of  the adoption of base model (Llama-chat vs Llama-notchat), evaluation method (GPT score vs. Bevertails moderation) or training method (LoRA vs Full fine-tuning). I think LoRA vs full fine-tuning is probably the reason leading to the discrepancy,  but I cannot be sure.  Totally fine if you don't have an answer to this comment.

I believe this paper has its value. Novel finding has been discovered and is important for subsequent research on attack dataset construction. I would like to see the authors rebuttal to decide my final rating.

**Relation To Broader Scientific Literature:**

Previous work (Qi et al.2023) first show that benign fine-tuning can compromise safety.

Previous work (He et al, 2024) show that the attack performance of benign fine-tuning can be increased by curating a subset among the avialble benign dataset (e.g., Alpaca)

This work show that the attack performance of benign fine-tuning can be further enhanced by an alternative data curation method. The authors conduct extensive empirical experiment to jusitfy that their data curation scheme is better than (He, et al, 2024).

**Theoretical Claims:**

Yes. The deduction in (1)- (5) is correct, though this theoretical claims are established by an existing paper.

---

> ### Author Rebuttal · Authors · 2025-04-01
>
> > Q1.Virus and some other missing relevant work
>
> Thanks for the suggestions. We have added all the work to our manuscript.
> > Q2.1 Additional experiment on GSM8k
> > Q2.2 Utility downgrade raises concern on fine-tuning
>
> Thanks for the great question.
>
> **Fine-tuning on GSM8K** We fine-tune LLaMA-2-7B-Chat on 100 samples selected via random, BA, and our method. The train-test split follows (He et al. 2024). Test accuracy is the percentage of correct answers, and HS is measured on the Hex-Phi dataset with GPT-4o as the judge.
>
> Method|HS|Acc
> -|-|-
> Un-fintuned|1.18|17.57%
> Random|2.50|20.45%
> BA|3.39|20.25%
> Ours|3.42|20.37%
>
> As seen, our method successfully unaligns the model, and fine-tuning slightly improves test accuracy, confirming its downstream benefits.
>
> **Why utility downgrades**: Unlike GSM8K, where utility is measured by test accuracy, Table 1 uses MT-Bench to evaluate general generation quality. A possible reason is that after fine-tuning, response lengths tend to be shorter, possibly to mimic the Dolly-style concise responses. Since MT-Bench relies on model-based evaluation, potential verbosity bias in the judge model[1] may contribute to the utility downgrade.
>
> Utility downgrade is also observed in Table 8 of [3]: Even fine-tuning on the full benign dataset leads to a slight drop. While scores slightly decrease, the generations are of similar high quality from human inspections.
> > Q3.Logistical flow for the attack method design
>
> Lines 147-161 outline our intuition: for an aligned LLM, clean and safe samples lie comfortably within the safety distribution, while harmful samples become “outlier” samples that lie outside the safety scope. We hypothesize that certain outlier samples within benign datasets may have a high potential to push LLM parameters toward harmful zones during fine-tuning. The following experiments are used to validate this intuition (Fig. 3&4).
> > Q4.What will happen to a dataset where most samples are short samples
>
> The short samples will be detected.
>
> We'd like to discuss a potential misunderstanding of the outlier samples. Our outlier detection is to find some outliers that are **away from the model's alignment space**, but not to detect relative outliers in the **given dataset** (see Reviewer RTza Q6). Thus, these short samples are selected, potentially because of their weird length compared to the training dataset of the aligned LLMs. Thus, regardless of the fine-tuning dataset's length distribution, the vanilla Self-Inf score always favors shorter samples. Therefore, the finding is not contradictory.
>
> Additionally, while short samples are effective, the revised Self-Inf-N, which normalizes length bias, is more useful in capturing 'harmful benign samples.' We welcome further discussion!
> > Q5. Advantages of our method to BA.
>
> - A novel outlier-based approach to breaking safety alignment.
> - A significantly more efficient method, requiring only one-third of BA's runtime.
> - No reliance on an anchor dataset.
> - More practical applications, e.g, continuous learning.
>
> Additionally, our primary goal is not to surpass BA in harmfulness. An average score >3 is sufficient to cause substantial safety degradation. Furthermore, since GPT-4o models serve as the judges, score variations might also exist.
> > Q6.The method seems to be a direct application of (Pruthi et al., 2020)
>
> Thanks. Please refer to Reviewer RTza Q2.
> > Q7.Naming issues(Malicious Fine-tuning)
>
> Thanks. We have revised our manuscript.
> > Q8.Lisa: Beavertails vs. RepNoise
>
> Yes, we use original Beavertails. The table below presents the Lisa defense results with repnoise_beavertail (same settings as Figure 8(a)), where refusal responses are used. The results show that the new Beavertail dataset effectively suppresses LLM harmfulness during fine-tuning.
> K1/K2 steps|HS
> -|-
> 2/18|1.52
> 6/14 |1.24
> 10/10|1.17
> 14/6 |1.02
> 18/2 |1.01
>
> This experiment made us reconsider our earlier claim that the bi-state optimization strategy has limited effectiveness in suppressing the harmfulness of benign samples. We have now included this experiment and revised our arguments accordingly in the updated manuscript.
> > Q9.Different observations in Lisa's Table 15.
>
> As discussed in Q8, Lisa’s performance improves significantly with repnoise_beavertail. We've updated our statement to reflect these findings, which partially align with Table 15 of Lisa's paper.
>
> However, key questions remain: Why does the alignment dataset influence Lisa’s performance? And how can we design better defenses against benign fine-tuning attacks? These might be interesting for future readers.
> > Q10.Why benign fine-tuning attack fails in Lisa's testbed.
>
> Lisa seems to fine-tune directly with BA-selected Alpaca samples, but the sample scores were computed using a LLaMA-Chat model in full fine-tuning mode. While BA's paper shows some transferability, a more direct approach would be replicating the scoring process with a LLaMA-NonChat model in LoRA mode.
>
> [1]https://arxiv.org/pdf/2310.10076

---

> > ### Comment · Reviewer_UXrg · 2025-04-02
> >
> > Hi authors,
> >
> > Thanks for the rebuttal. Most of my concerns are addressed. However, I still have several comments after reading the rebuttal:
> >
> > Q2.1, the results look fine. A regret is that Self-Inf-N basically share the same performance with BA. I read your answer for Q5., and understand the main contribution Self-Inf-N.
> >
> > Q.3 I still think the motivation of the design of Self-Inf-N is weak after reading your answer. Even though the outlier is over the aligned model, i.e., shorter answer can be more easily regarded as an outlier. However, this still can be a coincidence and does not form a good logistical flow. For example, let say the aligned model (i.e., chat model) is aligned by mostly short answers, then understandably the long answers can be the outlier. In this case, the self-inf-N method will select the long answers, which still contradict your own finding "Influence of Samples with Short Token Lengths." in Page 4.
> >
> > It seems that the fundamental issue of the method design is that the method is not based on a logistical thinking (or a basic conjecture) but start with some pure experiment trials. Yes, with coincidence you can get it right, however, you never know how and why the method actually work.
> >
> > With that said, other concerns are sufficiently addressed. I think most other part of the papers are fine. Novel finding  without covered by existing literature has been proposed. The new method indeed eliminate the use of anchor dataset in BA, which is good.  I will support this paper acceptance, and will actively participate in the reviewer-AC discussion phase. However, due to the remain concern. I insist to keep the borderline score.
> >
> > I guess you have another round to reply. Feel free to challenge me if you think what I said is wrong.
> >
> > **I saw the author's reponse**, and I appreciate the additional clarifications. I think this paper can be accepted as long as the clarification is included in the updated paper. I will actively participate in the reviewer-AC discussion and champion acceptance of this paper.

---

> > > ### Author Response · Authors · 2025-04-03
> > >
> > > Thank you very much for your encouraging words and for recognizing the main contributions of our work. We would like to take this valuable final rebuttal opportunity to further elaborate on the two insightful points you raised.
> > >
> > > > Q2.1, A regret is that Self-Inf-N basically shares the same performance with BA.
> > >
> > > We truly appreciate your recognition of the main contribution of Self-Inf-N. We would like to take this chance to highlight two additional points that we believe further demonstrate the uniqueness and practicality of our approach:
> > >
> > > **No Reliance on Anchor Dataset**: As you noted, our attack method achieves comparable performance to BA without relying on any additional anchor dataset.
> > >
> > > **Unique Advantages under Practical Scenarios**: Beyond the standard evaluation, Self-Inf-N also exhibits distinct advantages under more realistic settings. One such setting is continuous learning that we mentioned in Sec 4.4. Specifically, fine-tuning on a small set of only 100 selected samples neither provides enough task-specific knowledge nor ensures good performance on the downstream task, which makes it easier to identify as a useless LLM and raises suspicions and potentially leading users to discontinue. Therefore, we aim to investigate whether the attack is still valid in the continuous fine-tuning setting, where the fine-tuning contains two stages (Stage 1: benign fine-tuning with selected 100 samples; Stage 2: Domain-specific Fine-tuning with Asclepius dataset).
> > >
> > > As we explored in the continuous setting, once the model is fine-tuned with samples found by our method, the harmfulness can persist during the following continual fine-tuning stages. We also evaluate the same setting with BA. However, their harmfulness diminished much faster in the second fine-tuning stage than ours under different learning rates used in the second stage. Specifically, the results (same setting as in Figure 6 (b)) are as follows,
> > >
> > > |Setting $\downarrow$|HS of BA| HS of Ours|
> > > |-|-|-|
> > > |w/o first stage|1.31|1.31|
> > > |lr=5E-6|2.13|3.39|
> > > |lr=8E-6|1.87|3.20|
> > > |lr=2E-5|1.62|2.78|
> > >
> > > Here, HS denotes the average harmfulness score across 11 categories in Hex-PHI. "w/o first stage" indicates the model is fine-tuned solely on the Asclepius QA dataset. These results illustrate that the harmfulness introduced by our method is more enduring than BA’s in a continual fine-tuning setup, further reinforcing its practical impact.
> > >
> > > > Q.3 The motivation of the design of Self-Inf-N is weak after reading your answer.
> > >
> > >
> > > Thank you for the insightful question.
> > >
> > > **Supporting Evidence for the Intuition**: Our intuition for selecting outlier samples in the benign dataset is supported by two recent, well-established works. Prior research [1] has shown that harmful samples are explicitly distinct from harmless ones in the models' hidden states, while [2] has demonstrated that fine-tuning LLMs on datasets containing more harmful samples tends to cause greater shifts in model embeddings. Together, these findings partially support our intuition that outlier samples tend to induce greater weight deviations. Therefore, we propose to use Self-Inf as an outlier detection method to pick the outlier samples in the benign dataset, which have greater potential in shifting model weights. We have incorporated a discussion of these two papers into our explanation of the intuition, which we believe will strengthen our argument.
> > >
> > > **Clarifying the Length Bias Discussion**: The discovery regarding short token lengths is indeed an interesting and unexpected finding. However, we regard it as a secondary insight rather than a core motivation. Our primary goal is to develop a method that reliably induces harmfulness in LLMs. The exploration of length bias (Section 3.3) led us to refine Self-Inf into Self-Inf-N, which addresses this bias and forms our final proposed method.
> > >
> > > **Response to the Hypothetical Scenario**: Your counterexample—where aligned models are predominantly trained with short answers—is a thoughtful and important one. While most current LLMs are aligned using relatively long, detailed conversations, your scenario points to a meaningful direction for future exploration. We agree that under such circumstances, long answers might be outliers, and the Self-Inf approach might select them. We plan to acknowledge this limitation in the final version of our paper and are grateful for your suggestion.
> > >
> > > Once again, thank you for your detailed comments and thoughtful suggestions throughout the review process. Your feedback has significantly helped us improve the clarity, rigor, and broader relevance of our work.
> > >
> > > [1] On prompt-driven safeguarding for large language models.
> > >
> > > [2] Vaccine: Perturbation-aware alignment for large language model.

---

### Official Review · Reviewer_RTza · 2025-03-13

**Overall Recommendation:** 3

**Summary:**

This submission investigates a vulnerability in the fine-tuning stage of large language models (LLMs), demonstrating that benign datasets can be exploited to compromise safety alignment. The authors examine this problem via the lens of outlier identification, using Self-Inf-N to discover and remove outlier samples from benign datasets that may compromise LLM safety during fine-tuning.

The principal contributions encompass: (1) Detection of detrimental samples within benign datasets via outlier identification; (2) Creation of Self-Inf-N, a normalized self-influence scoring methodology that mitigates length bias in outlier detection; (3) Empirical findings indicating that fine-tuning on merely 100 outlier samples identified by Self-Inf-N can jeopardize the safety alignment of large language models; (4) Illustration of the attack's transferability across various model architectures and sizes; (5) Evaluation of the attack's efficacy in real-world contexts.

**Claims And Evidence:**

The claims made in the submission are supported by evidence, but there are some limitations that undermine the submission's significance.

The primary claim that "fine-tuning LLMs on 100 outlier samples selected by Self-Inf-N in the benign datasets severely compromises LLM safety alignment" is demonstrated across several models. The core research problem "Could seemingly harmless, benign samples be exploited to further undermine LLM’s safety alignment?" is developed based on prior work [1]. The authors' contribution of using outlier detection to identify the most problematic samples is incremental.

The evidence for the effectiveness of Self-Inf-N over vanilla Self-Inf is convincing, but this technical improvement does not sufficiently advance the field's understanding of LLM safety vulnerabilities.

Ref:
[1] Fine-tuning aligned language models compromises safety, even when users do not intend to! In ICLR, 2024.

**Essential References Not Discussed:**

N.A.

**Experimental Designs Or Analyses:**

The experimental designs and analyses in the submission are generally well-designed but have two limitations.

The main experiments on fine-tuning LLMs with outlier samples are heavily biased toward Llama-2 models, which are no longer representative of current LLMs. The field has evolved rapidly, with newer models incorporating more sophisticated safety alignment techniques that may be more resistant to the attack described in this paper.

The submission also lack experiments with closed-source fine-tuning services. While the authors mention in the introduction that "fine-tuning service providers (e.g., OpenAI Platform) might refuse the request for fine-tuning models on harmful datasets," they do not test whether their benign outlier samples could bypass the safety filters of these commercial platforms.

**Methods And Evaluation Criteria:**

The proposed method and metrics are suitable for the research issue.

The Self-Inf-N method enhances existing gradient-based impact estimating approaches by resolving a significant shortcoming (length bias) seen in the standard Self-Inf approach.

The evaluation criteria for safety uses the comprehensive and established benchmark. "GPT-4 as a judge" for assessing harmfulness is a standard metric for evaluating the harmfulness of the responses.

The experimental design employs various baselines and conducts ablation studies on many hyperparameters, offering a comprehensive evaluation of the attack's efficacy.

**Other Comments Or Suggestions:**

N.A.

**Other Strengths And Weaknesses:**

N.A.

**Questions For Authors:**

1. Have you tested Self-Inf-N on close-sourced models like GPT-4o?
2. How do you differentiate your contribution from [1]?
3. Given that most commercial LLMs are equipped with extensive safety evaluations, how would the proposed method realistically impact deployed systems?
4. Have you compared Self-Inf-N with other outlier detection approaches beyond gradient-based influence estimation, such as density-based or deep learning approaches for anomaly detection?


Ref:

[1] Fine-tuning aligned language models compromises safety, even when users do not intend to! In ICLR, 2024.

**Relation To Broader Scientific Literature:**

The core finding of this submission has already been discussed in [1]. While the authors introduce a new method for identifying the most problematic samples (Self-Inf-N), this represents a refinement rather than a fundamental advance in our understanding of LLM vulnerabilities.

The authors acknowledge prior work on malicious fine-tuning and benign fine-tuning compromising safety alignment but do not sufficiently differentiate their contribution from these existing works.

Ref:

[1] Fine-tuning aligned language models compromises safety, even when users do not intend to! In ICLR, 2024.

**Theoretical Claims:**

The paper does not make extensive theoretical claims or provide formal proofs, as it is primarily an empirical study.

---

> ### Author Rebuttal · Authors · 2025-04-01
>
> > Q1.The difference between our paper and [1]
>
> [1] raises an important observation: even fine-tuning on benign samples can lead to a certain degree of safety degradation. However, despite this initial investigation, two key questions remain unexplored:
>
> - The increase in harmfulness is quite limited and occurs only in certain categories (see Figure 1 in [1]).
> - Why does safety degradation happen? [1] attributes this phenomenon to "catastrophic forgetting" on the utility-oriented dataset but does not delve into deeper investigations.
>
> To address these questions, we take a step further by investigating whether a specific subset of the benign dataset contributes disproportionately to safety degradation. We approach this from a novel outlier detection perspective. We believe that the insights gained from our outlier detection analysis (as outlined in Q2) will enhance the community’s understanding of LLM vulnerabilities.
> > Q2.Technical improvement does not sufficiently advance the understanding of LLM safety vulnerabilities.
>
> **Technical improvement.** While Self-Inf is a well-established method [2,3], we are **the first to adapt it for benign fine-tuning attacks**. Importantly, our proposed Self-Inf-N introduces non-trivial refinements(mitigating length bias), which lead to significantly stronger attacks on LLMs (see Figure 4).
>
> The core objective of our paper is to uncover vulnerabilities in LLMs during fine-tuning. We believe our contributions offer meaningful insights for the community:
>
> - Short samples can severely degrade safety alignment (Figure 3; Section 3.3).
> - Self-Inf-N, a refined outlier detection method, improves attack effectiveness (Figure 4).
> - Strong empirical results show that Self-Inf-N misaligns LLMs across model sizes and architectures, and remains effective in continuous learning and data poisoning scenarios (Figure 4, Table 1, Figure 5).
>
> > Q3.The main experiments are biased toward Llama-2 models
>
> Thanks for the great question. We also evaluate our method with other more advanced methods like Qwen2-7b-instruct. The following table shows the results.
> Method|HS|
> -|-|
> Random Selection|1.63
> BA|3.05
> Ours|3.22
>
> It is shown that, our method still successfully leads the fine-tuned Qwen model to a safety degradation problem. We have added the experiments to our updated manuscript.
>
> > Q4.Lack experiments with closed-source models.
>
> Thank you for the great question. Our method requires gradient access, which is not feasible for closed-source models. However, we evaluate the transferability of samples selected using LLaMA-2-7B-Chat to GPT-3.5 and GPT-4. The results show limited impact, likely due to (1) large architectural differences between the two model series and (2) (potential) stronger safety mechanisms in closed models. We believe this is a valuable direction for future research.
> > Q5.How does the proposed method impact deployed systems?
>
> **Escaping existing safety moderations.** Firstly, our method can successfully escape the moderation tools (Sec 4.6), while the harmful samples would be easily detected.
>
> **Benign Fine-tuning breaks safety** Our attack method is effective for the popular fine-tuning-as-a-service setting for the deployed LLM: when the user uploads the selected "seemingly" benign samples, the safety of the fine-tuned LLM would be broken.
> > Q6.Comparisons with other outlier detection methods.
>
> **Results with DBScan** We conducted a simple experiment using DBSCAN, applying it to the last-layer hidden representations to identify 100 samples whose embeddings differ most from the overall distribution. The results on the Dolly dataset are shown in the table below. As observed, DBSCAN demonstrates significantly lower effectiveness compared to our method.
>
> Method|HS|
> -|-|
> DBScan|1.62
> Ours|3.71
>
> **Why do density-based and DL approaches fail?**  We believe the key difference lies in the objective: rather than identifying relative outliers **within the dataset**, our goal is to find samples that most strongly **influence the deviation of the aligned model’s parameters**. In other words, we intend to find samples that are most "shocked" (influential) to the aligned model $f_{\theta}$. Density-based and DL approaches, however, are not suitable for our setting as they mainly focus on finding relative outlier samples in the **given dataset**.
>
> **Why use (Pruthi et al., 2020)?** We chose the method from Pruthi et al. (2020) due to its proven effectiveness and ease of use in LLMs, as supported by recent work in instruction tuning data selection [2] and outlier analysis [3]. Additionally, compared to other gradient-based influence estimation methods (e.g., Data-inf), it offers significantly better memory efficiency.
>
> [1]Benign Samples Matter! Fine-tuning On Outlier Benign Samples Severely Breaks Safety
>
> [2]Less: Selecting influential data for targeted instruction tuning
>
> [3]Outlier gradient analysis: Efficiently improving deep learning model performance via hessian-free influence functions.

---

### Official Review · Reviewer_Bq3d · 2025-03-13

**Overall Recommendation:** 3

**Summary:**

This paper puts forth the idea that fine-tuning can compromise the safety of large language models. In particular, they leverage existing work from the field of outlier detection to exploit these data points in benign datasets and then demonstrate that fine-tuning on these examples (which are still, by construction, benign) trains a resultant model that is more likely to be harmful or non safety aligned.

**Claims And Evidence:**

Both scenarios 1 (continuous learning) and 2 (data poisoning) are well motivated as practical extensions of the proposed methodology.

The section on safety mitigation is well received. However, is there a reason why the authors only used API-based detection tools? Many model users also have access to model-based guardrails such as [LlamaGuard](https://arxiv.org/abs/2312.06674), [GraniteGuardian](https://arxiv.org/abs/2412.07724), or [WildGuard](https://arxiv.org/abs/2406.18495). It would have been good to corroborate any findings from the API based detection setups with one or more of these model-based guardrails.

**Essential References Not Discussed:**

N/A

**Experimental Designs Or Analyses:**

The experimental design makes sense overall.

**Methods And Evaluation Criteria:**

The harmfulness evaluation has a few issues.

1. The harmfulness of the resultant models is evaluated through a single dataset of 330 samples. This is quite limited, and begs the question why the authors didn't consider other evaluation benchmarks. In particular, this benchmark is rather small and may not capture the full extent of harm benchmarks. Why not consider other commonly used safety benchmarks, such as the [BBQ dataset](https://aclanthology.org/2022.findings-acl.165/) or [ToxicChat](https://arxiv.org/abs/2310.17389) or [XSTest](https://arxiv.org/abs/2308.01263). Note that the last dataset, XSTest, might be particularly useful for this instance given its motivation.
2. The harmfulness evaluation comes from an auxiliary model (namely, GPT-4). The authors do not provide the specific prompt that is used here, so it is hard to judge. Additionally, there is no corroboration of the judge model's effectiveness - either through a small human validation or comparing to other state of the art harm detection models such as [LlamaGuard](https://arxiv.org/abs/2312.06674), [GraniteGuardian](https://arxiv.org/abs/2412.07724), or [WildGuard](https://arxiv.org/abs/2406.18495)

Also small note that the main text says GPT-4 is used, but Appendix Section A ays GPT-4o is used.

**Other Comments Or Suggestions:**

See above.

**Other Strengths And Weaknesses:**

A strength of the paper is the breadth of models considered in the experiments, with most if not all of them being fully open source which is a big positive.

**Questions For Authors:**

See above.

**Relation To Broader Scientific Literature:**

The authors do a good job of relating their current setup to all of the previous work on malicious fine-tuning and how it may compromise the safety alignment of LLMs.

Specifically, they do well to provide links to all of these similar works and also continue to reference them throughout the paper.

**Theoretical Claims:**

The theoretical grounding in influence functions via data influence estimation is well presented. Additional background regarding outlier detection is also provided.

---

> ### Author Rebuttal · Authors · 2025-04-01
>
> > Q1.Why using API-based detection tools?
>
> Thank you for the insightful question. **We totally agree that the suggested evaluation model can further enhance the robustness of our analysis, hence we have already included it in our updated results.**
>
> Method|LlamaGuard|GraniteGuard|WildGuard
> -|-|-|-
> Harmful Dataset|9, 91|6, 94|5, 95
> Ours|100, 0| 100, 0|100, 0
>
> Each cell shows two values: the number of detected safe samples and unsafe samples, respectively. Our method clearly outperforms the harmful dataset.
>
> We also respectfully argue that API-based and model-based detection tools are fundamentally similar, as API-based tools also rely on **underlying models** trained on large, crowd-labeled datasets for general-purpose safety detection (similar to how ChatGPT operates). We chose API-based tools due to their widespread adoption in both academic research[6] and commercial applications[1].
> > Q2.Why not considering other safety benchmarks, e.g., BBQ, ToxicChat, and XSTest.
>
> **We fully agree that including additional benchmarks leads to a more comprehensive evaluation. Hence we have incorporated both the BBQ and XSTest datasets to further evaluate our method on LLaMA-2-7B-Chat**. For XSTest, we followed the original paper's setup and used GPT-4 to classify the model’s responses to unsafe queries into three categories: full_compliance, full_refusal, and partial_refusal.
>
> Method|full_compliance|full_refusal|partial_refusal
> -|-|-|-
> w/o finetuned|14|182|4
> BA|130|63|7
> Ours|146|50|4|
>
> For BBQ, we computed the bias scores on both ambiguous and disambiguated subsets (higher scores indicate more bias):
>
> Method|Bias Score (ambig)|Bias Score (disambig)
> -|-|-|
> w/o finetuned|18.66|39.17
> BA|47.50|46.51
> Ours|46.95|46.47
>
> As seen, LLM fined-tuned with our method exhibits a more compliance to the harmful query prompts in XSTest, and a more biased outputs in BBQ, compared to BA and un-finetuned version.
>
> **The limitations of ToxiChat, BBQ, and XSTest**
> - ToxiChat primarily evaluates toxicity detection tools (e.g., OpenAI Moderation API), which falls outside the scope of our work.
> - BBQ focuses on bias in models' generations, which is **just one dimension** of safety.
> - XSTest is indeed a good benchmark, which contains ten safety categories, , but mainly used to study over-refusal behaviors. Compared to HEx-PHI, it has fewer unsafe prompts (200) and covers safety categories less comprehensively.
>
>
> **Why we use Hex-PHI?** Hex-PHI is a well-established benchmark for evaluating LLM's safety. It is grounded in the 'exhaustive lists of prohibited use cases found in **Meta’s Llama-2 usage policy and OpenAI’s usage policy**.'[2], covering a diverse range of safety categories such as **illegal activity, privacy violations, and child exploitation**. It has also been adopted in several recent works [3,4,5], making it a strong choice for evaluating LLM safety in our context.
>
>
>
> > Q3.Prompt for the judge model
>
> As noted in Appendix A, 'The prompt provided to the judge model is identical to that used in [2]'. Please refer to [2] for more details.
> > Q4.Corroboration of the judge model's effectiveness
>
> Thank you for the thoughtful question. **Following your suggestion, we further evaluate the generated 330 responses on Hex-PhI using additional guard models: LlamaGuard, GraniteGuard, and WildGuard.**
>
> Method|LlamaGuard|GraniteGuard|WildGuard
> -|-|-|-
> w/o Finetuned|321 safe+9 Unsafe |324 Safe+6 Unsafe|328 safe+2 Unsafe
> BA|103 safe+227 Unsafe|63 Safe+267 Unsafe|105 Safe+225 Unsafe
> Ours|100 safe+230 unsafe|70 safe+260 unsafe|104 safe+226 Unsafe
>
> These results lead to the same conclusion that our method leads to significantly more unsafe responses. However, through manual inspection, we identified two limitations of these tools: (1) High false negative rate – many harmful outputs are incorrectly labeled as safe. (2) Limited granularity – guard models are trained on predefined categories and may not detect nuanced harms (e.g., economic harm in HEx-PHI is hard to be detected). Therefore, we believe LLM-as-a-judge remains a more reliable evaluation method for our setting.
>
> Using LLM-as-a-judge is a widely accepted practice for safety evaluation, as adopted by several well-established works [2,3]. (Reviewer UXrg and RTza also noted, GPT-4 as a judge for assessing harmfulness is a standard metric).
>
> > Q5.GPT-4 vs. GPT-4o
>
> Thanks for your detailed review. That was indeed a typo, we used GPT-4o in our experiments.
>
> [1]https://medium.com/jigsaw/reducing-toxicity-in-large-language-models-with-perspective-api-c31c39b7a4d7
>
> [2]Fine-tuning Aligned Language Models Compromises Safety, Even When Users Do Not Intend To! (ICLR 2024)
>
> [3]Safety alignment should be made more than just a few tokens deep. (ICLR 2025)
>
> [4]Artprompt: Ascii art-based jailbreak attacks against aligned llms. (ACL 2024)
>
> [5]SafeDecoding: Defending against Jailbreak Attacks via Safety-Aware Decoding. (ACL 2024)
>
> [6]Bias and fairness in large language models: A survey

---

### Official Review · Reviewer_cFgx · 2025-03-14

**Overall Recommendation:** 3

**Summary:**

This paper investigates a vulnerability in the fine-tuning stage of LLMs, where even benign datasets can lead to a significant increase in the harmfulness of LLM outputs. The authors propose a novel attack method, Self-Inf-N, which identifies and selects outlier samples from benign datasets to fine-tune LLMs, thereby compromising their safety alignment. Specifically, the proposed attack exhibits high transferability across different architectures and remains effective in practical scenarios. Furthermore, most existing mitigation strategies fail to defend against this attack, highlighting the need for more robust alignment safeguards. The experiments demonstrate the effectiveness and transferability of the attack.

**Claims And Evidence:**

The claims made in the paper are generally supported by clear and convincing evidence. However, there are some points could be further clarified:
- The paper discusses practical scenarios like continuous learning and data poisoning, but the experiments are somewhat limited in scope.
- The authors suggest that benign samples could avoid toxicity detection which is trivial as the data are benign samples but the toxicity detection tools are designed for harmful training data. Is there any design suggestion in mitigating safety issue during fine-tuning?

**Essential References Not Discussed:**

Although using a different approach, there are some concurrent works that could be discussed in a future revision:

- Hsiung et al. "Your Task May Vary: A Systematic Understanding of Alignment and Safety Degradation when Fine-tuning LLMs" (2024)
- Mu et al. "Stealthy Jailbreak Attacks on Large Language Models via Benign Data Mirroring" (NAACL 2025)

**Experimental Designs Or Analyses:**

The experimental designs and analyses are generally sound.

**Methods And Evaluation Criteria:**

The proposed methods and evaluation criteria are appropriate.

**Other Comments Or Suggestions:**

none

**Other Strengths And Weaknesses:**

**Strengths**
- This paper addresses a critical and underexplored vulnerability in LLM fine-tuning.
- The proposed Self-Inf-N method is novel and effectively mitigates the length bias in the original Self-Inf method.
- The extensive experiments demonstrate the effectiveness and transferability of the attack across multiple LLMs.

**Weaknesses**
- This paper did not discuss any corresponding defense mechanism in addressing the potential threats.

**Questions For Authors:**

Please refer to the weaknesses

**Relation To Broader Scientific Literature:**

This paper could contribute to making LLM safe and mitigate fine-tuning vulnerabilities.

**Theoretical Claims:**

The paper does not present any theoretical proofs.

---

> ### Author Rebuttal · Authors · 2025-04-01
>
> > Q1. The paper discusses practical scenarios like continuous learning and data poisoning, but the experiments are somewhat limited in scope.
>
> Thank you for the insightful question. The continuous learning and data poisoning settings represent our preliminary exploration into how the proposed method might realistically threaten LLM alignment in practice. In particular, we incorporated practical considerations such as a real-world medical downstream task (Asclepius) and experiments with low poisoning ratios (see Section 4.4.2). While we acknowledge that our current work cannot exhaustively cover all practical scenarios, we hope these initial investigations can serve as a foundation for future work in this important area.
>
> > Q2. The authors suggest that benign samples could avoid toxicity detection which is trivial as the data are benign samples but the toxicity detection tools are designed for harmful training data. Is there any design suggestion in mitigating safety issue during fine-tuning?
>
> Thank you for raising this important point.
>
> **Any Mitigation Strategies During Fine-tuning?** We have explored **additional mitigation strategies** in Section 4.6, including both *data augmentation* and *fine-tuning-stage defense* methods. Specifically, we evaluate whether incorporating extra alignment samples can mitigate harmfulness, and we also test advanced fine-tuning defenses such as LISA [3]. Empirical results show that augmenting the fine-tuning dataset with safety samples from the Bianchi dataset would effectively suppress the harmfulness.
>
> **Why do we incorporate toxicity detection?** We agree that it is expected for benign samples to bypass toxicity detectors, and we include this analysis primarily to highlight **the stealthiness of such attacks in practice**.
>
> > Q3. Although using a different approach, there are some concurrent works that could be discussed in a future revision:
>
> Thanks for these suggestions. We have included the suggested papers in our updated manuscript!
>
> > Q4. This paper did not discuss any corresponding defense mechanism in addressing the potential threats.
>
> In Section 4.6, we identify a promising mitigation strategy: augmenting the fine-tuning dataset with a subset of safety-aligned data from Bianchi et al., which appears to reduce the harmfulness of LLM outputs. We also discuss the limitations of existing defense methods and emphasize the need for more advanced techniques specifically designed to address benign-sample-based alignment attacks. We hope this work can help motivate further research in this critical direction.
>
> [1] Hsiung et al. "Your Task May Vary: A Systematic Understanding of Alignment and Safety Degradation when Fine-tuning LLMs" (2024)
>
> [2] Mu et al. "Stealthy Jailbreak Attacks on Large Language Models via Benign Data Mirroring" (NAACL 2025)
>
> [3] Huang T, Hu S, Ilhan F, et al. Lisa: Lazy safety alignment for large language models against harmful fine-tuning attack[J]. Advances in Neural Information Processing Systems, 2024, 37: 104521-104555.

---

### Decision · Program_Chairs · 2025-05-01

**Decision:**

Accept (spotlight poster)

**Comment:**

This paper investigates a vulnerability in the fine-tuning stage of LLMs, where even benign datasets can lead to a significant increase in the harmfulness of LLM outputs. The authors propose a novel attack method, Self-Inf-N, which identifies and selects outlier samples from benign datasets to fine-tune LLMs, thereby compromising their safety alignment. Specifically, the proposed attack exhibits high transferability across different architectures and remains effective in practical scenarios. Furthermore, most existing mitigation strategies fail to defend against this attack, highlighting the need for more robust alignment safeguards. The experiments demonstrate the effectiveness and transferability of the attack.

This paper has many strengths. For example, the paper is easy to read. The performance is better than existing SOTA solution by (He et al, 2024). The attack is tested against SOTA defense against harmful fine-tuning. The authors show that their attacks are effective even though defense such as safeInstrc or Lisa is adopted. The evaluation is really impressive because some of the defense just appear very recently. The code is well organized with runnable script. This make it easier to reproduce the results. The finding that short answer benign sampels sriously downgrade safety but fine-ting only on them compromise utility is novel. I also agree the authors explanation with shallow alignment hypothesis.

While the reviewers had some concerns about evaluations and claims, the authors did a particularly good job in their rebuttal. Therefore, all of us have agreed to accept this paper for publication! Please include the additional discussion in the next version.